# Prokaryotic Expression of Phospho*enol*pyruvate Carboxylase Fragments from Peanut and Analysis of Osmotic Stress Tolerance of Recombinant Strains

**DOI:** 10.3390/plants10020365

**Published:** 2021-02-14

**Authors:** Jiaqi Tu, Lanlan Feng, Yanbin Hong, Qiuyun Liu, Xia Huang, Yin Li

**Affiliations:** 1Guangdong Key Laboratory of Plant Resources, School of Life Sciences, Sun Yat-sen University, Guangzhou 510275, China; cathytu@cyagen.com (J.T.); fengll@link.cuhk.edu.hk (L.F.); lsslqy@mail.sysu.edu.cn (Q.L.); huangxia@mail.sysu.edu.cn (X.H.); 2Crops Research Institute, Guangdong Academy of Agricultural Sciences, Guangzhou 510640, China; hongyanbin@gdaas.cn

**Keywords:** phospho*enol*pyruvate carboxylase, *Arachis hypogaea* L., *Escherichia coli*, recombinant, osmotic stress

## Abstract

Phospho*enol*pyruvate carboxylase (PEPC) is a ubiquitous cytosolic enzyme that catalyzes the irreversible β-carboxylation of phospho*enol*pyruvate (PEP) in presence of HCO_3_^−^ to produce oxaloacetate (OAA) during carbon fixation and photosynthesis. It is well accepted that *PEPC* genes are expressed in plants upon stress. *PEPC* also supports the biosynthesis of biocompatible osmolytes in many plant species under osmotic stress. There are five isoforms of *PEPC* found in peanut (*Arachis hypogaea* L.), namely, *AhPEPC1*, *AhPEPC2*, *AhPEPC3*, *AhPEPC4,* and *AhPEPC5*. Quantitative real-time polymerase chain reaction (qRT-PCR) analysis revealed that the gene expression patterns of these *AhPEPC* genes were different in mature seeds, stems, roots, flowers, and leaves. The expression of all the plant type PEPC (*PTPC*s) (*AhPEPC1, AhPEPC2, AhPEPC3,* and *AhPEPC4*) was relatively high in roots, while the bacterial type PEPC *(BTPC)* (*AhPEPC5*) showed a remarkable expression level in flowers. Principal component analysis (PCA) result showed that *AhPEPC3* and *AhPEPC4* are correlated with each other, indicating comparatively associations with roots, and *AhPEPC5* have a very close relationship with flowers. In order to investigate the function of these *AhPEPC*s, the fragments of these five *AhPEPC* cDNA were cloned and expressed in *Escherichia coli* (*E. coli*). The recombinant proteins contained a conserved domain with a histidine site, which is important for enzyme catalysis. Results showed that protein fragments of *AhPEPC1*, *AhPEPC2,* and *AhPEPC5* had remarkable expression levels in *E. coli*. These three recombinant strains were more sensitive at pH 9.0, and recombinant strains carrying *AhPEPC2* and *AhPEPC5* fragments exhibited more growth than the control strain with the presence of PEG6000. Our findings showed that the expression of the *AhPEPC* fragments may enhance the resistance of transformed *E. coli* to osmotic stress.

## 1. Introduction

Phospho*enol*pyruvate carboxylase (PEPC; EC 4.1.1.31) is a key enzyme in plant metabolic pathways such as photosynthesis. PEPC is a tightly regulated cytosolic enzyme in cytoplasm, catalyzing the irreversible β-carboxylation of PEP in presence of HCO_3_^−^ to produce oxaloacetate (OAA) and inorganic phosphorous (Pi) with magnesium (Mg^2+^) ions as a cofactor [1,2]. PEPC is abundantly present in the plant kingdom as well as green algae and cyanobacteria, most of the archaea, and non-photosynthetic bacteria [2]. On the basis of function and phylogeny, the PEPCs can be divided into two subfamilies, the plant type PEPC (PTPC) and the bacterial type PEPC (BTPC). *PTPC* genes encode proteins of size 100–110 kDa with a conserved N-terminal seryl-phosphorylation site and a critical C terminal tetrapeptide QNTG. It is reported that all PTPCs originated from a common ancestor [3,4,5,6]. *BTPC* genes encode larger proteins (116–118 kDa) of partial similarity with PTPC, which has a prokaryotic-like (R/K) NTG C-terminal tetrapeptide [5,6]. Both PTPCs and BTPCs have the important domains required for catalytic and substrate binding, the BTPCs resemble the bacterial PEPCs rather than the common plant PEPCs. BTPC proteins do not possess N-terminal seryl-phosphorylation domain of PTPCs [7,8].

The role and regulation for PEPCs in plants have been extensively investigated due to their important role in carbon fixation and photosynthesis [1,2]. PEPCs in plants are involved in the synthesis of storage compounds by the anaplerotic reaction. These storage compounds enter the tricarboxylic acid cycle (TCA) with intermediates such as pyruvate and phospho*enol*pyruvate (PEP) involve in a variety of biosynthetic pathways [1,9]. In addition to the photosynthesis, PEPC also plays important role in carbon–nitrogen interactions, maintaining cellular pH, seed formation and germination, fruit maturation, control of stomatal movements, supplying carbon for symbiotic N_2_-fixation in root nodules of leguminous plants, and regulates the mechanism of stress tolerance in plants [10,11,12,13]. 

Peanut (*Arachis hypogaea* L.) is an economically important oilseed legume distributed across more than 115 countries with 26 million hectares of area [14]. Previously, five *PEPC* genes (*AhPEPC1*, *AhPEPC2*, *AhPEPC3*, *AhPEPC4,* and *AhPEPC5*) were isolated from peanut and the complexity of the peanut *PEPC* gene family was analyzed using phylogenetic relationship, gene structure, and chromosome mapping in peanut [15,16]. It was reported that *AhPEPC1*, *AhPEPC2*, *AhPEPC3,* and *AhPEPC4* are typical PTPCs, and *AhPEPC5* is a BTPC [13,15,16]. The interesting finding is that the expression level of *AhPEPCs* is lower in the high oil yielding cultivated peanut varieties than in the cultivated peanut varieties with normal oil yield except *AhPEPC2* [15]. 

The *AhPEPC*s’ gene expressions vary depending on the tissue, organ, and stage of development [13,16]. Pan et al. [13] analyzed the expression patterns of *AhPEPC1*–*5* in peanut cultivar E11 under abiotic stress (cold, salt, and drought) conditions and observed that the expression of *AhPEPC3* showed high expression under abiotic stress, whereas the expression of *AhPEPC1* and *AhPEPC2* showed moderate expression. *AhPEPC1*, *AhPEPC4,* and *AhPEPC5* were down-regulated under salt stress, and *AhPEPC4* and *AhPEPC5* were down-regulated under drought stress, indicating that peanut PEPCs might be playing different roles upon various abiotic stresses. However, there are few reports on the function of these five genes using the ectopic expression. Therefore, in this study, five *AhPEPC* coding sequences from cultivated peanut were analyzed, and cDNA segments were cloned in *Escherichia coli*. These *AhPEPC* coding sequences containing one of the five conserved domains with a histidine site were transformed into *E. coli* for prokaryotic expression analysis. The recombinant *E. coli* growth analysis showed that *AhPEPC1*, *AhPEPC2,* and *AhPEPC5* had brought resistance of transformed *E. coli* to osmotic stress, but the recombinant strains turned out to be more sensitive to basic condition. These results provide molecular evidence for the function of the *AhPEPC* genes in response to stress tolerance in plants.

## 2. Results

### 2.1. Expression Patterns of AhPEPCs in Peanut

Quantitative real-time PCR (qRT-PCR) was used to monitor the expression patterns of the five *AhPEPC* genes in different peanut tissues. The relative expression levels of *AhPEP*Cs are shown in Figure 1. The results showed that the expression of *AhPEPC1* was relatively high in flowers, roots, and leaves when compared with mature seeds and stems. The expression levels of *AhPEPC2* in roots, flowers, leaves, and stems were relatively high, but low in seeds. The *AhPEPC3* was relatively high in roots and flowers, lower in stems, leaves, and seeds, while *AhPEPC4* showed much higher accumulation in roots and flowers than in leaves, stems, and seeds. The *AhPEPC5* showed high expression in flowers, moderate in leaves, and low expression in other tissues, especially in seeds. The expression of all the *PTPC*s (*AhPEPC1*, *AhPEPC2*, *AhPEPC3,* and *AhPEPC4*) was relatively high in roots, while the *BTPC* (*AhPEPC5*) showed a remarkable expression level in flowers.

The five *PEPC* genes exhibited different gene expression patterns, suggesting that these genes may have different roles in peanut. In order to find interrelationships among *AhPEPC*s gene expression levels in different organs tested, a principal component analysis (PCA) was performed (Figure 2). The two principal components PC1 and PC2 covered approximately 88.3% cumulative proportion of the total variance. Based on the distribution of different samples in the PCA biplot and contributions of variables, *AhPEPC3* and *AhPEPC4* are correlated with each other, indicating comparative associations with roots; moreover, there is a very close relationship between *AhPEPC5* and flowers.

### 2.2. Cloning of AhPEPC cDNA Fragments and Construction of Prokaryotic Expression Vectors

The cDNA fragments of *AhPEPC1* (688 bp), *AhPEPC2* (574 bp), *AhPEPC3* (538 bp), *AhPEPC4* (529 bp), and *AhPEPC5* (568 bp) were amplified by PCR (Appendix A) based on the cDNA sequences of *AhPEPC1* (EU391629), *AhPEPC2* (FJ222240), *AhPEPC3* (FJ222826), *AhPEPC4* (FJ222827), and *AhPEPC5* (FJ222828) from NCBI database. The fragments encoded peptides contained one of the important conserved domains (VLTAHPT) necessary for enzyme activity, harboring the catalytic site of a histidine residue [3] (Figure 3). The five *AhPEPC* fragments obtained are consistent with predictions, which encodes 24.64, 21.01, 19.14, 18.89, 20.24 kDa fraction of *AhPEPC1*–*5* protein, respectively. The five fragments were inserted into the prokaryotic expression vector pET-28a with a 6 × His tag, respectively, and then transformed into *E. coli* DH5α and confirmed by PCR (Appendix A) and DNA sequencing. The corresponding DNA fragments were detected in agarose gel electrophoresis, and the cloned fragments are identical to the sequences in NCBI database.

### 2.3. Expression of AhPEPC Fragments in E. coli

The recombinant expression vectors carrying *AhPEPC* fragments were transformed into Rossetta (DE3) competent cells to optimize the expression conditions for the efficient production of recombinant proteins. The SDS-PAGE analysis results (Figure 4) showed that three recombinant strains with transgenes for the fragment of *AhPEPC*s produced the peptides of the expected size (25–33 kDa) with remarkable efficiency. The size of predicted peptides composed of 6xHis tag fused to the fragment of *AhPEPC* protein was 30.03 kDa for *AhPEPC1*, *AhPEPC1*, 26.40 kDa for *AhPEPC2,* and 25.63 kDa for *AhPEPC5*. The amounts of isopropyl β-d-thiogalactopyranoside (IPTG)-induced peptides were gradually increased with increasing time of induction.

Total bacterial proteins of the three strains were tested by the SDS-PAGE and Western blot to study whether the strains contain His-tag (Figure 5). We observed that the protein band of the size of the target protein increased significantly after induction. Western blotting results showed that the recombinant peptides of *AhPEPC1*, *AhPEPC2,* and *AhPEPC5* contained His-tag, and the immunoblotting of His-tag was not observed in the control.

### 2.4. Effect of Osmotic Stress on the Growth of Recombinant Strains 

To do a quick assessment of the peanut PEPC function, ectopic expression of *AhPEPC* peptides in *E. coli* were determined to evaluate whether the peptides affect the growth of *E. coli* and represent a function against osmotic stress [17,18]. In controlled trials, there was no apparent difference in cell growth between the *AhPEPC* fragments carrying strains and the control strains (Appendix A). The effects of *AhPEPC1*, *AhPEPC2,* and *AhPEPC5* recombinant peptides were checked by polyethylene glycol (PEG6000) treatment. As shown in Figure 5, after 6 h of IPTG induction, there was no significant difference in cell growth between the *AhPEPC2*, *AhPEPC5*-expressing strains, and the vector control. However, the *AhPEPC1*-expressing strains that produced a larger protein showed growth inhibition when compared with control (Figure 6a). The treatment of 10% PEG6000 inhibited the growth of all recombinant strains and the vector control strains, however, *AhPEPC2* expressing strains showed more tolerance to osmotic stress than other strains (Figure 6b). 

When the concentration of PEG6000 reached 20%, as up to 12 h the *AhPEPC2* expressing strains showed more tolerance, and the growth of *AhPEPC1*-expressing strains were inhibited to a lesser extent compared to the vector control (Figure 6c). These results showed that recombinant strains overexpressing *AhPEPC*s improved osmotic tolerance.

### 2.5. Acid/base Tolerance Test of Recombinant Strains

*AhPEPC1*, *AhPEPC2,* and *AhPEPC5*-expressing strains were cultured in media with pH range 3.0–9.0. In culture media of pH 3.0 to pH 7.0, the strains showed similar growth patterns to the vector control (data not shown). However, when the pH was increased to pH 9.0, the growth of these three recombinant strains was severely inhibited, which indicated that recombinant strains are sensitive at basic pH (Figure 6d).

## 3. Discussion

Plant PEPC is a key enzyme in photosynthesis, seed development [19,20,21,22,23], nutrient accumulation and metabolism [16,24,25], and abiotic stress adaptation [1,26,27,28,29]. One of the important seed crops, peanut, has five *PEPC* genes that can regulate the metabolic process of fatty acid and protein biosynthesis in seeds [16]. In the present study, we further characterized these peanut *PEPC*s using the prokaryotic expression technique. The results offered experimental evidence for the relation of *PEPC* in peanut to osmotic stress. 

Previous studies [13,15] reported gene expression patterns of the five *PEPC* in roots, stems, seeds, and leaves while no report in flowers was available. In our qRT-PCR assays, we found the distinctive expression of the *BTPC AhPEPC5* in flowers. The expression patterns of the five *AhPEPC* genes are strikingly different between normal and high-oil varieties but do not offer any obvious clues to the function of the *AhPEPC* genes [13,15]. However, it was presumed that lower-level expression of *AhPEPC*s facilitates the flow of PEP into lipid in high-oil peanut variety [15]. The analysis of results during seed development (10–60 d after pegging) also presumed that lower-level expression of *AhPEPC* genes and the decrease in PEPC activity may be in favor of lipid accumulation in peanut seeds [16]. In the present study, we analyzed the *AhPEPC*s expression in peanut cultivar Shanyou 523, which is a normal-oil variety, and the results showed differential gene expression patterns. The plant materials used in this research were not from the same developmental stage as those in previous studies. The PCA results supported that *AhPEPC4* might have an important function in roots, and the correlation of *AhPEPC4* and *AhPEPC3* implies that they may have similar functions. The expression patterns of five *PEPC* in flowers of cultivated peanut have not been investigated before. It is notable that all five *PEPC*s were more abundant in flowers than in leaves. The *AhPEPC5* showed a different expression pattern as it expressed much higher in flowers when compared to other organs tested. There is a remarkable connection between *AhPEPC5* and flowers in the PCA biplot, indicating that this bacterial type *AhPEPC* may also play an important role in flower development.

Prokaryotic expression by using *E. coli* as a host is widely used to validate the protein function. Overexpression of plant-derived PEPC gene in *E. coli* can enhance tolerance when exposed to high temperature, salinity, dehydration, or methyl viologen [17], and recombinant expression of *PEPC* gene fragment can enhance the regulation of protein and lipid accumulation [30], suggesting that plant-derived PEPC may have a function in *E. coli*. In the present study, we had expressed fragments of five *AhPEPC*s containing one conserved domain (VLTAHPT) with an histidine site, not include other highly conserved motifs which found in plant type of PEPC [3,31]. A previous study revealed that the PEPC activity of recombinant *E. coli* harboring a peptide from *Chlamydomonas reinhardtii* has elevated 24% higher than wild type [30]. The protein fragments in the present study correspond to similar region of the peptide. In PEPC, there are only two conserved histidines, and site-directed mutagenesis of these histidines has identified their essential role in catalytic activity [32,33]. These two conserved histidines together with other residues are important for catalytic activity and the inhibitory effector aspartate [33]. We had expressed *AhPEPC* fragments containing one of the conserved domains VLTAHPT and the first histidine site. Site-directed mutagenesis and three-dimensional structure analysis suggested that this histidine site involves in the carboxylation of PEP with bicarbonate anion to form oxaloacetate [32,34]. In plants, PEPC converts PEP to OAA by carboxylation, subsequently malate and eventually lead to the synthesis and accumulation of osmotically active compounds and improves drought tolerance in plants [18,35,36]. Pan et al. [13] have found that PEPC genes in peanuts showed differential transcription abundance in leaves during cold, salt, and drought stresses, suggesting functions of regulation in responses to abiotic stresses. Our investigations did not find the recombinant strains showing resistance to high or low temperature, salinity, and oxidative stress (data not shown), These incomplete peptides could not have full activity but may compete for a substrate with native enzymes, and the ectopic expression of *AhPEPC2* and *AhPEPC5* fragments may have conferred osmotic stress resistance to the recombinant strains to a certain extent. Further investigation is needed to study whether the whole or larger fragment of the protein shows better effect.

Park et al. [37] expressed a PEPC protein from marine bacteria *Oceanimonas smirnovii* in *E. coli* and found that the purified PEPC protein maintains enzymatic activity in basic pH 9.0–10.0, but almost no activity at acidic pH 5.0. PEPC up-regulation may increase organic acid synthesis, including malate and citrate, and PEPC activities in root tips of plants are much higher at basic pH [38]. PEPC also participate in proton sink through the regulation of malate and pyruvate metabolism [39]. Cheng et al. [18] found that recombinant strain expressing a PEPC protein from C4 halophyte *Suaeda aralocaspica* could adapt to a wider range of pH at 5.0 and 9.0 than that of control, but we did not find similar results in this study. In the present study, the recombinant strains showed more sensitivity to basic conditions and showed growth inhibition under basic pH 9.0. It is possible that the incomplete *AhPEPC* peptides had led to substrate competition but no corresponding proton transfer reaction occurs, which leads to the recombinant strains being more sensitive to the basic environment.

## 4. Materials and Methods

### 4.1. Plant Materials, E. coli Strains, and Plasmids

A peanut cultivar ‘Shanyou 523’ was provided by the Crop Research Institute, Guangdong Academy of Agricultural Sciences (Guangzhou, China). Seeds were planted in 2.5-L soil-containing pots and grown till the flower stage in a greenhouse, at 25 °C with a 14 h light and 10 h dark cycle. Plant material used for RNA isolation was collected from 50 days old peanut plants at full flowering stage. Root tissues were collected from the young growing part of the main roots. Leaf and stem tissues were collected from the third compound leaf below the terminal bud and the corresponding stem segment of the sampled leaves and the mature seeds were collected about 50 days after pegging. Collected samples were immediately frozen and stored at −80 °C until use. Three biological replicates were used for each treatment. *E**. coli* strain DH5 α and Rossetta (DE3) was from Tiangen Biotech (Beijing, China) Co. Ltd. The vector pET-28a was procured from Beijing Dingguo Changsheng Biotechnology Co. Ltd. (Beijing, China). 

### 4.2. AhPEPC Fragments Cloning and Construction of Expression Vectors

Total RNA was extracted using the RNAprep Pure Plant Kit (Polysaccharides and Polyphenolics-rich) (Tiangen, Beijing, China), cDNA was synthesized according to the instruction in PrimeScript™ 1st Strand cDNA Synthesis Kit (TaKaRa, Kyoto, Japan). Primer pairs for the five PEPCs in peanut were designed based on the cDNA sequences available in NCBI database, these were *AhPEPC1* (accession number: EU391629), *AhPEPC2* (accession number: FJ222240), *AhPEPC3* (accession number: FJ222826), *AhPEPC4* (accession number: FJ222827), and *AhPEPC5* (accession number: FJ222828). DNAMAN software was used for multiple sequence alignment. The gene fragments were amplified with gene-specific primers (Appendix A), an initiation codon (ATG) and an *EcoR* I site were incorporated in the forward primers. Gene fragments were digested with restriction enzymes and ligated into a prokaryotic expression vector (pET28a, Beijing Dingguo Changsheng Biotechnology Co. Ltd., Beijing, China) to generate the expression plasmid pET28a-*AhPEPCs*. PCR reaction was carried out as follows: initial denaturation (94 °C, 2 min), 32 cycles (94 °C for 15 s, 58 °C for 30 s, 72 °C for 60 s), and final extension of 72 °C for 10 min. The resultant *AhPEPC* fragments generated from PCR were inserted into the pET-28a vector to construct the prokaryotic expression vectors. The heat shock method was used to transform the plasmids into *E. coli* DH5α. The positive clones were selected by PCR and then verified by sequencing.

### 4.3. Quantitative Gene Expression Analysis

Expression patterns of *AhPEPC*s in peanut tissues were investigated using qRT-PCR, and total RNA was extracted from roots, stems, leaves, and mature seeds of flowering peanut plant. Subsequently, cDNA was synthesized. Transcript-specific primers were designed and the peanut *18S rRNA* was taken as the reference gene to calculate the expression levels [15]. Primers were presented in Appendix A. The qRT-PCR reaction mixtures were prepared using a Premix Ex *Taq*TM (TaKaRa, Kyoto, Japan), reactions were performed as the manufacturer’s instructions on an ABI StepOne Plus system [40]. Each 20 μL reaction contained 2.0 μL template, 10 μL 2×Premix Ex *Taq*TM, and 0.8 μL (10 μM) of each primer. qRT-PCR was conducted at initial denaturation 95 °C, 3 min, then 40 cycles of 95 °C, 10 s, 60 °C, 15 s, and 72 °C, 20 s. Relative expression was calculated using the 2^−ΔΔCT^ method [41]. Each measurement was carried out in triplicate with three biological replicates.

### 4.4. Prokaryotic Expression and Western Blot Analysis

The prokaryotic expression strain *E. coli* Rossetta (DE3) was transformed with the correctly sequenced plasmid. For prokaryotic expression analysis, the strains were cultured on a rotary shaker with incubator at 37 °C and 200 rpm speed. During log phase, an isopropyl β-d-thiogalactopyranoside (IPTG) concentration of 0.4 mM was used to induce the fusion protein expression at 37 °C for 6 h. The bacterial suspension was harvested by centrifugation for 0, 2, 4, and 6 h, and the fusion protein was separated by 12% sodium dodecyl sulfate-polyacrylamide gel electrophoresis (SDS-PAGE) and visualized using Coomassie brilliant blue (CBB) staining. Protein samples were transferred to an Immobilon Polyvinylidene Fluoride (PVDF) membrane (Millipore, Burlington, MA, USA), 5% skim milk powder was used to block the membrane. Membranes were treated with His-Tag mouse monoclonal antibody (Proteintech) and a secondary antibody of horseradish peroxidase (HRP) conjugated affinipure goat anti-mouse IgG (H + L) (Proteintech), and detected using Chemiluminescence Ecl Detection Kit (Millipore, Burlington, MA, USA).

### 4.5. Growth and Stress Tolerance Assay

The strains Rossetta: pET-28a-*AhPEPC*s (recombinant) and Rossetta: pET-28a (control) were inoculated into 5 mL fresh Luria-Bertani (LB) medium and incubated at 37 °C overnight. Cultures were diluted at proportion of 1:100 into fresh LB and grown at 37 °C for 2.5 h. When the optical density at 600 nm (OD600) reached 0.6, IPTG was added to 0.4 mM to induce protein expression for 12 h. The growth rate of recombinant and control strains was measured by recording the OD600 at 2 h interval. To perform stress tolerance assay, the recombinant strains and control strain were incubated; when the OD600 reached 0.6, IPTG was added to 0.4 mM to induce protein expression for 6 h. The OD600 values were measured and the cultures were diluted to 0.5 OD600, then 200 μL of each culture were inoculated into 20 mL fresh LB medium containing 10%, 15%, 20% polyethylene glycol (PEG) 6000 for osmotic stress assay. For acid and base tolerance test, the pH value of the culture was adjusted to 3.0, 5.0, 7.0, 9.0. All cultures were incubated on a shaking incubator at 220 rpm overnight at 37 °C. Growth rates were measured by recording the OD600 every 2 h. Three biological repeats were carried out for each treatment.

### 4.6. Statistical Analysis and PCA Analysis

Data were analyzed using SPSS Statistics 26.0. Statistical analyses were performed by Student’s *t*-test. All data were expressed as means ± standard deviation (SD). *p* < 0.05 and *p* < 0.01 are considered statistically significant. A PCA analysis was conducted for the expression levels of the five *PEPC* genes using the R statistical package (https://www.r-project.org/ accessed on 5 February 2021).

## Figures and Tables

**Figure 1 plants-10-00365-f001:**
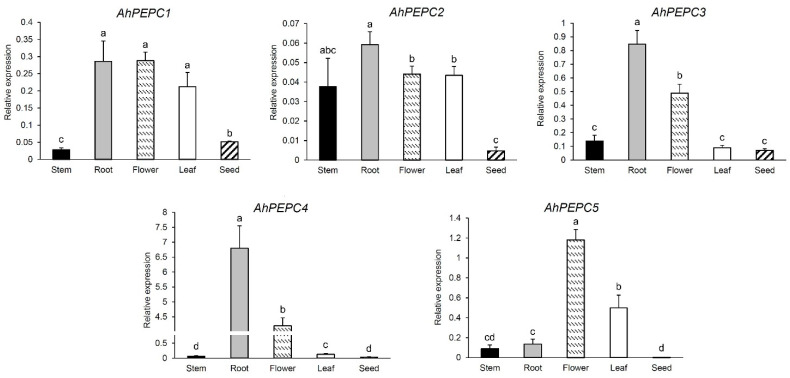
Expression patterns of five *AhPEPC* genes using qRT-PCR in different organs of peanut cultivar Shanyou 523. The graphs illustrate means± SD of three replicates. The bars not sharing the same letter have a statistically significant difference in gene expression level (*p* < 0.05).

**Figure 2 plants-10-00365-f002:**
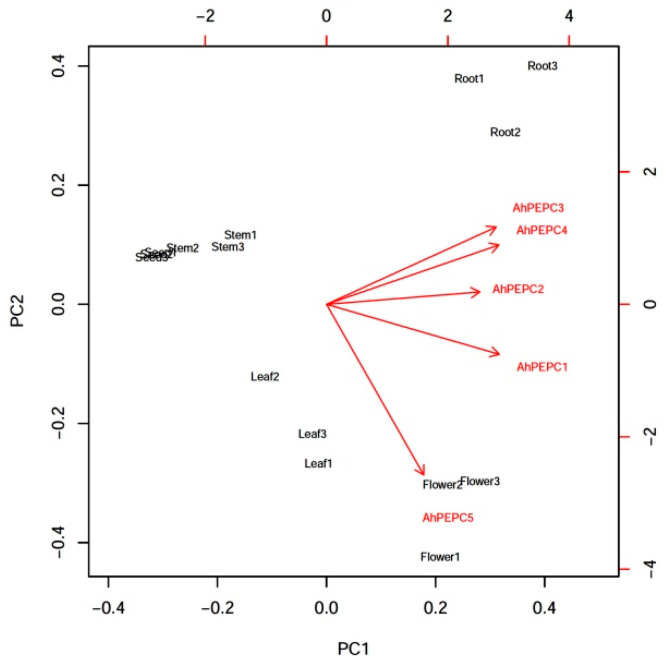
The results of principal component analysis (PCA) of the expression levels of the five phospho*enol*pyruvate carboxylase (PEPC) genes in different peanut organs. A distribution of the samples analyzed showed by a biplot, horizontal-axes represent principal component 1 (PC1) and vertical-axes represent principal component 2 (PC2). Arrows indicate the directions of *AhPEPC*s gene expression levels. PC1 and PC2 explain 68.7% and 19.6% of the data variances, respectively.

**Figure 3 plants-10-00365-f003:**
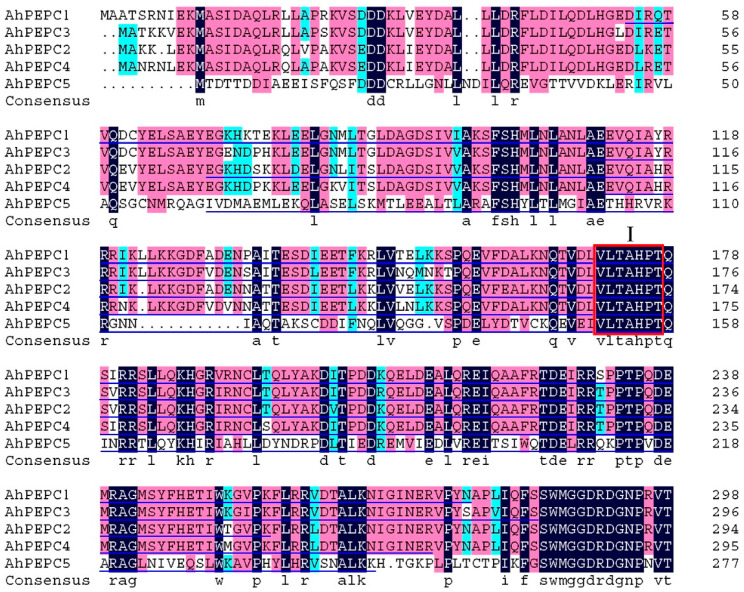
Alignment of part of five *AhPEPC* protein sequences from peanut. The underlines indicate the *AhPEPC* peptides encoded by the cloned cDNA fragments used for recombinant expression, and the box I domain is one of the conserved domains important for enzyme catalysis [3].

**Figure 4 plants-10-00365-f004:**
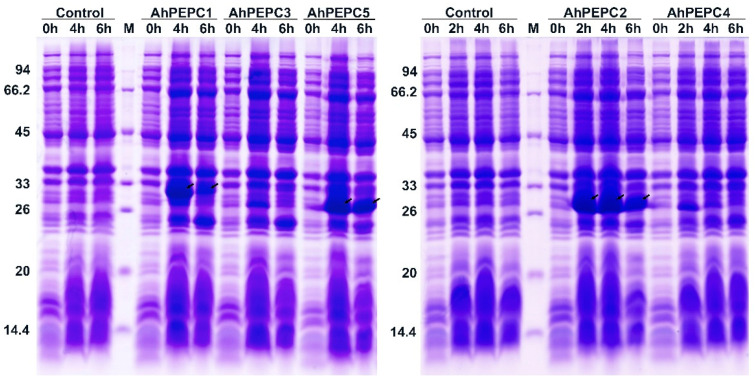
SDS-PAGE analysis of recombinant AhPEPC peptides expression in Escherichia coli Rossetta (DE3) with Coomassie blue staining. The pET-28a harboring stain was used as vector control. Arrows indicate the band of AhPEPC His-tag fusion peptides.

**Figure 5 plants-10-00365-f005:**
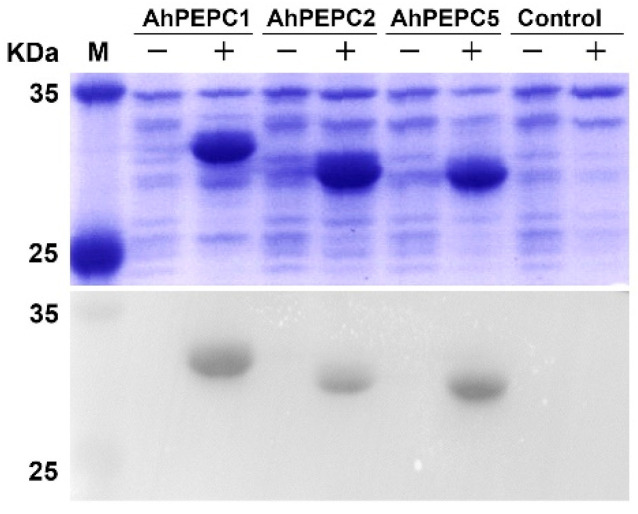
Western blot analysis of the expression of the recombinant *AhPEPC* with anti-His antibody. Specific protein bands were detected in three induced recombinant pET-28a-*AhPEPC* strains. M: Prestained protein molecular weight marker; +: Induced with 0.4 mM isopropyl β-d-thiogalactopyranoside (IPTG); −: Control.

**Figure 6 plants-10-00365-f006:**
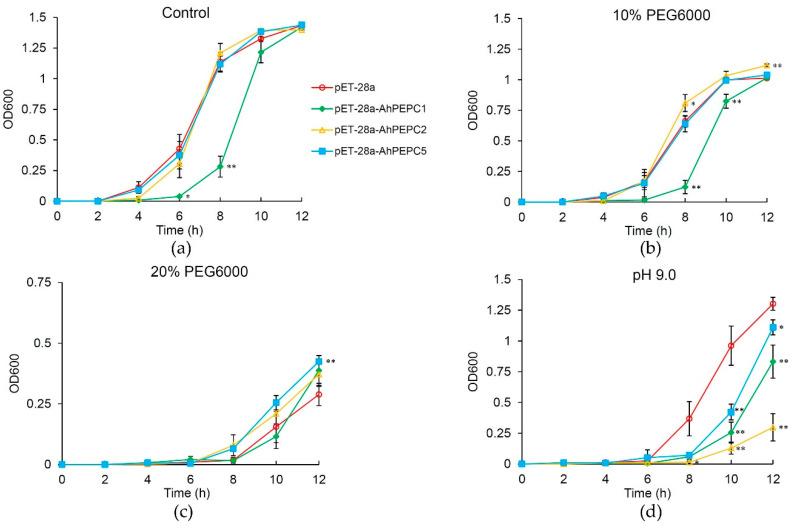
Time course of growth of the recombinant and control *E. coli* strains and the control strain under various abiotic stresses. (**a**) Culture under non-stressed condition after 6 h IPTG induction; (**b**) 10% PEG6000; (**c**) 20% PEG6000; (**d**) pH 9.0. The cultures were sampled at every 2 h interval for 12 h. Data points represent the means ± SD of three replicates. * *p* < 0.05, ** *p* < 0.01.

## Data Availability

The data presented in this study are available on request from the corresponding author.

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
