# Peer review of "Prokaryotic Expression of Phosphoenolpyruvate Carboxylase Fragments from Peanut and Analysis of Osmotic Stress Tolerance of Recombinant Strains"

_plants, 2021, doi:10.3390/plants10020365_

Round 1

Reviewer 1 Report

The authors studied the gene expression level of five isoforms of PEPC in different tissues of one peanut cultivar. Furthermore, protein fragments containing catalytic site of genes were cloned and expressed in E.coli. and osmotic stress survival assays of three recombinant strains were reported. Overall, the manuscript is well written. I have a few points which need to be addressed.

  1. Please add statistical analyses to the data presented in Fig. 1 and Fig. 5.
  2. Abiotic stress test was not performed for recombinant strains containing fragments of PEPC3 and PEPC4, which is due to the low expression in E.coli after induction? what is the explanation for this? is there any hypotheses/speculation to explain the differential responses of PEPC1, 2 and 5 to osmotic stress? The above points should be addressed in the discussion section. To me, it is a pity that no test was done for the PEPC4 as it shows the highest expression level in plant tissues as compared to other isoforms.

Reviewer 2 Report

Comments the manuscript:

I recommend the manuscript “Procaryotic Expression of Phosphoenolpyruvate Carboxylase Gene Fragments from Peanut and Analysis of Osmotic Stress Tolerance of Recombinant Strains” by Jiaqi Tu, Lanlan Feng, Yanbin Hong, Qiuyun Liu, Xia Huang and Yin Li for MDPI Plants. The manuscript concerns a plant that is industrially important and the genes being the object of author’s research are related to the basic life processes of the cell and participate in plant development and plant responses to stress factors.

However the manuscript requires a profound linguistic correction. I have marked some linguistic corrections for the first four pages of the manuscript (JP-reviewed-manuscript.pdf). However, these are not all the necessary corrections, and many paragraphs require rewriting. The remaining part of the manuscript also requires extensive linguistic revision. I would suggest the authors to ask a colleague fluent in English for help or use the service of a professional translational company to improve the language of the manuscript. 

Apart from the language corrections, I have reservations about the characteristics of plant material, interpretation of gene expression results, and the enzymatic activity of tested peptides, which are presented below in four points (Main remarks 1 - 4).

Main remarks to the Manuscript by Jiaqi Tu et al.:

1

The authors analyzed expression pattern of all five peanut PEPC genes in stem, root, flower, leaf and mature seeds. In my opinion the plant material used for RNA isolation have to be described more precisely especially that in the introduction authors write that PEPC genes expression vary in different organs depending on developmental stage. Even in the case of seeds, the term 'mature seeds' is imprecise, and in the case of other organs there is no information on the stage of development of the plant from which the material was taken and the degree of development of the organ. In the discussion the authors compare their results of the PEPC genes expression analysis with previously reported results for two different varieties of peanut, but do not provide any comment concerning differences in the result. Do they result from using varieties of different oil content in seeds (what is mentioned in the introduction), or differences in organs or plant development?

2.

I cannot agree with conclusions on expression patterns in the discussion (lanes 192 – 200):

A – AhPEPC2 expression in roots is very low and the gene contribute to less than 1% of all plant-type PEPC transcripts, so there is no basis for the author’s statement that this gene “might be playing an important role in root tissue” (lanes 193 – 195).

B – Expression of AhPEPC2 in flowers is lower than in roots and the same as in leaves so this genes is not “preferentially expressed in flowers” (lane 196, 197). Moreover its expression in flowers is more than 100 times lower than the other PEPC genes.

C – The authors are right that the expression of AhPEPC5 in flower is much higher than its expression in other organs, so this gene reveals specialization. But it is worth to point that its expression in flowers is lower than that of AhPEP4, so PEPC activity in peanut flower depends more on AhPEPC4 than on AhPEPC5.

I propose that the authors analyze AhPEPC genes expression results in such a way as to identify the genes that supply the majority of PEPC transcripts in given organ.

3.

The authors used recombinant E. coli strains with transgenes encoding peanut PEPC peptide fragments to analyze how high expression of these transgenes affects bacterial growth and stress response. However, the authors did not demonstrate experimentally that the peptides have PEPC enzymatic activity and did not justify it in the Discussion. The references [17], [36] and [37] concern works where the whole, native PEPC protein was analyzed. Moreover the work of Park [37] concerns PEPC from bacteria and not plant, similar as references [31] and [32]. Only in the work of Tian [29] the enzymatic activity was demonstrated for a fragment of plant PEPC produced in E. coli, but the work concern plant bacterial-type PEPC from Chlamydomonas reinhardtii. Even here the authors do not write whether the part of AhPEPC peptides they use in their work correspond to the same or similar region of PEPC as peptide used in the work of Tian [29]. The site directed mutagenesis and protein structure modeling which suggest that histidine from the conserved region present in the peptides used by authors directly participates in carboxylation reaction are not a proof that the peptides possess enzymatic activity, especially that there are evidences that plant PEPC activity depends on protein tertiary structure, [17]: “The PEPC has been postulated to exist as an active homo-tetramer or less active homodimer or inactive monomer/multimeric aggregate”. It seems that such a possibility is signaled at the end of the discussion (lane 237, 238), where authors propose that incomplete AhPEPEC peptides compete for substrate with native, active enzymes. However, the authors cannot postulate that such mechanism only occurs in this experiment and does not affect other studies, including the study of osmotic stress resistance.

Summarizing, the authors should present more arguments that the peptides they are analyzing have enzymatic activity in bacteria. When there is no such data, please explain how the study of an inactive peptide can reveal its functions, what is the molecular mechanism of its action in bacteria. Please also explain or correct the statement that “proteins (…) fragments may have partial function … “ (lane 223 – 225).

4.

In the title of the manuscript, as well as in many places in the text, it is stated that fragments of the PEPC peanut genes have been cloned or that fragments of these genes have been used for peptide synthesis (lanes 2, 79, 80, 109, 110, 113, 257, and others) which is not a true since the description in the materials and methods shows that the cDNA fragments were cloned and further used for the expression constructs assembly. The terms such as gene, cDNA, and transcript are not interchangeable. The other inaccuracies that make it difficult to understand the manuscript are:

1 – “peptide expression” (lane 125, 137, 145, 224 and others) – peptide is a product of gene expression

2 – Fig. 2. Alignment of five AhPEPC protein sequences from peanut (lane 127) – the presented peptides are only part of the whole AhPEPC proteins.

3 – The authors should provide readers with data on the size of the native protein or the native protein fraction they used in their research.

Reviewer 3 Report

The article entitled " Procaryotic Expression of Phosphoenolpyruvate Carboxylase Gene Fragments from Peanut and Analysis of Osmotic Stress Tolerance of Recombinant Strains " refers to the analysis of the five isoforms of PEPC enzymes in peanut. Authors determined the expression of individual isoforms in selected tissues, showing its diversity. Additionally, recombinant E. coli lines were obtained, determining their growth rate under the influence of abiotic stress. The article is written clearly and understandable. The introduction contains all the information that is needed to introduce the reader to the subject. The figures are elaborate and well described.

However, I have a few comments:

- methodology - how long after sowing the material was collected, in what stage of development were plants?

- AhPEPC2 expression is very low, what were the Ct values ​​in qPCR? I am afraid whether these differences are not due to insufficient sensitivity of the method and not from real differences in the level of the transcript

- the discussion compares the expression of the studied genes with the results obtained by another team. There is a lack of information on how such differences may arise, and how these varieties differ from each other?

- it is wondering why the development of AhPEPC1 recombinant bacteria is so strongly retarded, it causes questionable conclusions about different development in analyzed conditions

- the development of recombinant batteries was significantly different only under the influence of pH = 9.0, the remaining conditions did not significantly influence the differences in the development of recombinant and control bacteria. In the discussion, it was mentioned that changes in the expression of PEPC genes were described in peanut under the influence of abiotic factors. But maybe the examination on the bacterial model is not correct? The only possible solution would be to check the expression of the studied genes in peanut under conditions of abiotic stress (e.g. a simple callus or protoplast model).

- Figure S1. why in PCR unspecific  products were amplified? Has it been checked what has been amplified?

- There are also unspecific products in the pictures of gels in the figure S2, it would be worth redisign the reaction parameters

Round 2

Reviewer 1 Report

All points have been addressed by authors, the manuscript can be accepted as publication in Plants.

Author Response

Thank you very much for your comments and valuable improvements on our paper. We appreciate the time and effort that you dedicated to providing feedback on our manuscript.

Reviewer 3 Report

Thank you for your answers. It is possible that at first I did not notice the basic purpose of the work, which, by the way, is not defined anywhere. 

Author Response

(The authors gave the same response as above.)
